# Altered stability of nuclear lamin-B marks the onset of aging in male *Drosophila*

Wei-Qi Lin[1], Zhen-Kai Ngian[1], Tong-Wey Koh[1], Chin-Tong Ong [1,2] *

1 Temasek Life Sciences Laboratory, National University of Singapore, Singapore, Singapore, 2 Department of Biological Sciences, National University of Singapore, Singapore, Singapore

* chintong@tll.org.sg

**Data Availability Statement:** The data that support the findings of this study are available as follow: https://figshare.com/articles/dataset/Altered_

## Abstract

Epigenetic alterations occur during aging, but it remains unclear what epigenetic features are associated with the onset of physiological decline in animals. Nuclear lamin-B forms the filamentous meshwork underneath the nuclear envelope, providing the structural scaffold necessary for genome organization and gene regulation. We found that reduced level of nuclear lamin-B protein coincides with the decline in locomotor activity and stress resistance in young adult male *Drosophila*. Ubiquitous *lamin-B* expression improves locomotor activity of the male flies at the expense of lower stress resistance and shorten lifespan. This observation suggests that tissue-specific expression of lamin-B may regulate different aspects of animal physiology during aging. To test this hypothesis, specific GAL-4 lines were used to drive the expression of *lamin-B* in specific neuronal populations and muscle tissues in male flies. Ectopic expression of *lamin-B* in the dopaminergic neurons within the protocerebral anterior medial region of the brain improves the locomotor activity of the male flies with little impact on their stress responses and lifespan. Interestingly, age-dependent decrease in the level of lamin-B protein is independent of its mRNA expression. Instead, cellular thermal shift assay showed that lamin-B and CP190 insulator protein undergo significant change in their solubility during aging. This suggests that the increased solubility of lamin-B protein may contribute to its reduced stability and degradation during aging.

## Introduction

Aging is associated with changes in epigenetic features such as DNA methylation, histone modifications and RNA splicing patterns [1–3]. However, it is not fully clear what epigenetic mechanisms are linked to or initiate the onset of physiological decline in animals. Nuclear lamins are intermediate filaments that underlie the inner nuclear envelope to provide structural scaffold for the nucleus [4]. They play important gene regulatory roles by forming highly repressive lamina-associated domains (LADs) through their interactions with distinct chromosome regions [5]. There are A- and B-types of lamins based on their structural features and expression patterns. Mutations in human *lamin-A* gene cause premature aging disorder called Hutchinson-Gilford progeria syndrome and many other age-related diseases collectively called laminopathies [6]. Loss of *lamin-B1* could also trigger cellular senescence [7, 8]. Like

stability_of_nuclear_lamin-B_marks_the_onset_
of_aging_in_male_Drosophila/19307036.

**Funding:** The work is supported by Temasek Life Sciences Laboratory core funding 3160. The funders had no role in study design, data collection and analysis, decision to publish, or preparation of the manuscript.

**Competing interests:** The authors have declared that no competing interests exist.

mammals, *Drosophila* possess both A- (also known as *lamin-C*) and B-type lamins (*lamin-B* or *Dm0*). In old flies, age-associated loss of lamin-B leads to de-repression of immune-response genes and retrotransposons in the fat bodies, which can cause systemic inflammation and gut hyperplasia [9, 10].

While nuclear lamins regulate aging in different animal models, it is unclear if they may be involved in triggering the onset of physiological decline in adult animals. Using *Drosophila* as a model, we found that gradual decline in the level of nuclear lamin-B protein correlates with the reduction in the locomotor activity and stress resistance of young male adult flies. Ectopic expression of lamin-B in the dopaminergic neurons improved the locomotor activity of transgenic male flies, suggesting that the level of lamin-B protein is crucial to maintain animal physiology during aging. Interestingly, the mRNA level of lamin-B remains unchanged throughout *Drosophila* lifespan. Instead, cellular thermal shift assay suggests that the decline in the nuclear lamin-B protein may be caused by its increased solubility during aging.

## Materials and methods

### Fly strains

The following lines were obtained from Bloomington Drosophila Stock Centre (ID number): *DJ667-Gal4* (#8171) and *UAS-LamB.GFP* (#7378). *Hsp70-Gal4* was from Dr Yu CAI; *TH-Gal4* and *R58E02-Gal4* (#41347) lines were from Dr Tong-Wey KOH.

The *Hsp70-* and *R58E02-Gal4* line were backcrossed at least ten times while other *Gal4* lines were backcrossed more than five times to $w^{1118}$. The heterozygous *Gal4*/+ backcrossed lines were then mated with homozygous *UAS-LamB.GFP* to obtain heterozygous *Gal4/ UAS-LamB.GFP* flies and their control siblings +/*UAS-LamB.GFP* for the different experiments (See S3A Fig in S1 File). Male flies were used for all experiments except from S2B to S2D Fig in S1 File where female flies were analysed.

### Immunoblot and antibodies used

Five heads and bodies dissected from adult flies under light $CO_2$ anaesthesia were homogenized with plastic pestle (Sigma, Cat#Z359947) in 100 μl of 1 x Laemmli buffer (62.5 mM Tris-HCl pH 6.8, 2% SDS, 8.3% glycerol, 0.01% bromophenol blue, 0.1M DTT). After boiling, 10 μl of heads or bodies lysate was resolved on SDS-polyacrylamide gel and transferred to PVDF membrane using standard protocol. After blocking with 5% milk in TBST (0.05% Tween-20), the membranes were probed with the following primary antibodies: 1:2000 anti-Lamin-Dm0 (lamin-B, DSHB, Cat#ADL67.10-s), 1:1000 anti-Lamin-C (lamin-A, DSHB, Cat# LC28.26), 1:10000 anti-CP190 and 1:1000 anti-CTCF [11], 1:5000 anti-histone H3 (Abcam) and 1:2000 β-tubulin (DSHB, Cat#E7-s). The blots were developed with SuperSignal™ West Pico PLUS Chemiluminescent Substrate (Thermo Scientific, Cat# 34578). Protein bands were quantified with Adobe Photoshop CS3 (Version 10.0) using either β-tubulin (Fig 1A and 1B; S1A, S1B, S2A and S2B Figs in S1 File) or total lysate (Fig 5C) as control. Statistical test was done using *t*-test in Excel (Microsoft) and the graphs were plotted with GraphPad Prism software (Version 8.0.1).

### Lifespan assay

Lifespan of male flies were monitored daily from the various starting time-points for different genotypes. Twenty males were housed in a vial and food was changed every two days. Deaths were recorded every day until all flies perished. For each genotype, a minimum of 120 flies were assessed. The lifespan for each fly was calculated to generate a lifespan curve for each

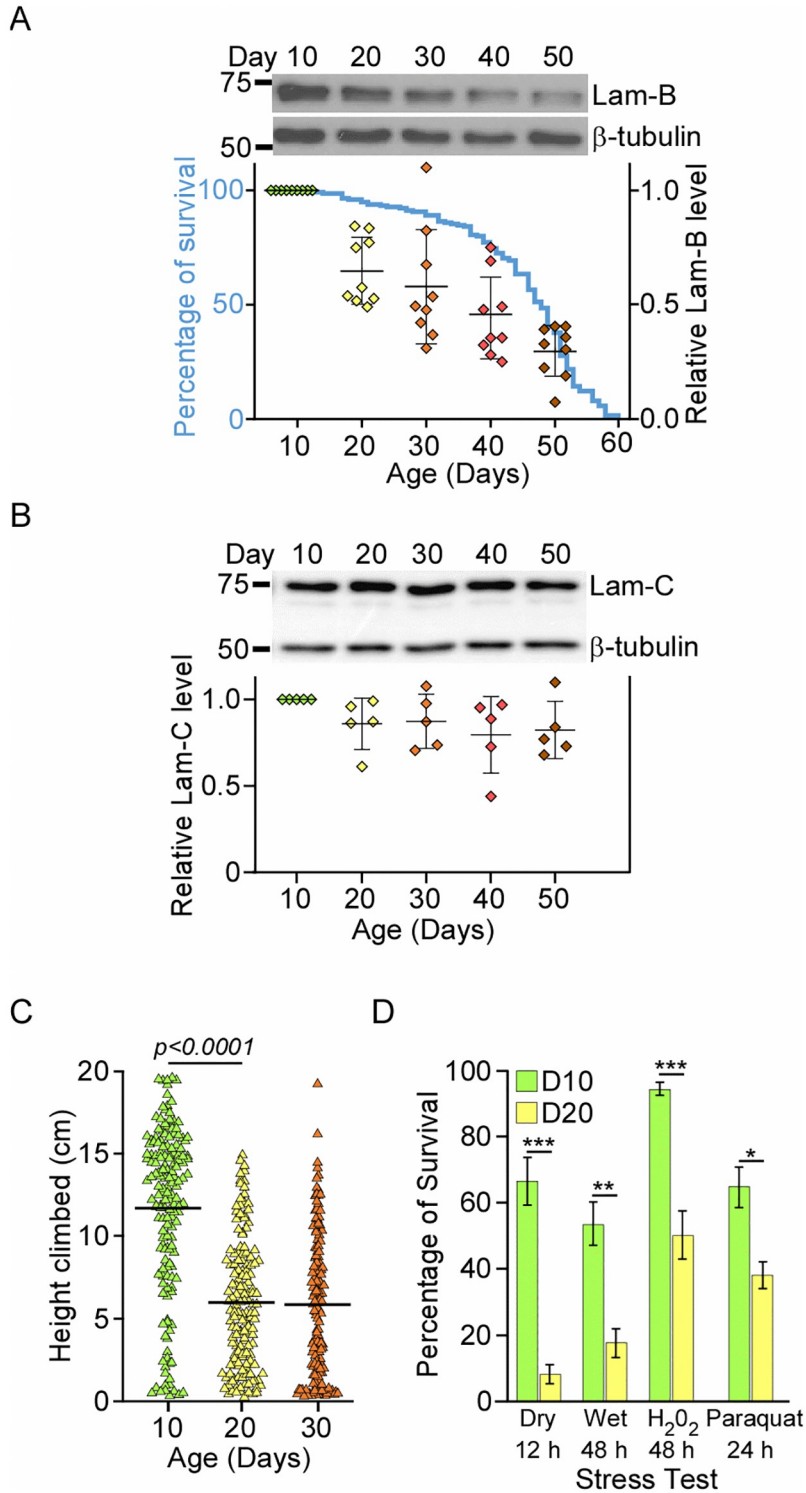

**Fig 1. Lamin-B protein marks the onset of physiological decline in young adult male *Drosophila*.** (A-B) Western blot and quantification of the head lysates from different ages across male $w^{1118}$ lifespan (blue, n = 180). Each biological replicate (diamond) has five heads. Data presented as level of (A) lamin-B and (B) lamin-C proteins relative to Day10 ± SD. (C) Locomotor activity of $w^{1118}$ males at different ages. Each triangle represents a fly and bar denotes mean. Mann-Whitney test. (D) Stress tests where flies were cultured in empty vial (dry), 1% agarose (wet), food with 1% $H_2O_2$ or 10 mM Paraquat. $^*$ $p < 0.002$, $^{**}$ $p = 0.0003$, $^{***}$ $p < 0.0001$.

genotype. Log-rank test (Prism) was performed to obtain a median lifespan and the corresponding *p*-value.

## Nuclear isolation protocol

Male flies (0.3mg) were homogenized with pestle in 300 μl buffer A1 (60 mM KCl, 15 mM NaCl, 4mM $MgCl_2$, 15 mM HEPES pH7.6, 0.5% NP-40, 0.5 mM DTT with proteinase inhibitors). After filtering with mira clothes, 50 μl of cellular lysates were collected as "Total" fraction. The lysates were then centrifuged at 4000 g for 5 min in cold and the supernatant was collected as "Cytoplasmic" fraction. The pellet was then washed three times with 0.5 ml of A1 buffer and harvested as "Nuclear" fraction. The different fractions were boiled in laemmli buffer and subjected to western blot analysis.

## Stress assays

For dry starvation, male and female flies were transferred into empty vials and monitored hourly between 6 to 12 hours. For wet starvation, flies were cultured in vials containing 1% agarose with the number of surviving flies recorded at 24 h and 48 h. For oxidative stress, flies were cultured in 10 ml of freshly made 1% sucrose and 1% agarose media that contained either 1% of hydrogen peroxide or 10 mM Paraquat (Sigma). Food with 1% of hydrogen peroxide was changed daily. Flies were monitored after 12 h, 24 h and 48 h. Stress test was conducted with 8–10 flies/vial and a total 88–139 male and 52–155 female flies/condition. The percentage of survival was plotted with appropriate statistical tests conducted (S1 Table in S1 File).

## Modified cellular thermal shift assay (CETSA)

CETSA measures the thermal stability of proteins under different conditions. For instance, cellular lysates exposed to different drugs are heated at various temperatures and centrifuged into different fractions [12]. Free proteins denatured by heating will precipitate into pellet whereas proteins protected by specific drugs remain soluble. We took advantage of the relatively stable and insoluble nature of lamin-B meshwork within the nuclear lamina to examine possible differences in their thermal stability between young and older male flies. Day 10 and Day 30 male flies were selected for this experiment as nuclear lamin-B showed significant reduction in the bodies only on Day 30 (S1B Fig in S1 File). Liquid nitrogen frozen bodies from Day10 and 30 male $w^{1118}$ flies were sieved to remove the heads and appendages using standard sized mesh (Scienceware, Cat#F37845-1000). 100 mg of fly bodies was grinded with 30 strokes of the Dounce homogenizer (Pestle A) (Sigma, Cat#D9938) in 5 ml of cold A1 buffer (60mM KCl, 15mM NaCl, 4mM $MgCl_2$, 15mM HEPES pH 7.6, 0.5% NP-40, 0.5mM DTT, and protease inhibitors including 1x Leupeptin, Pepstatin A, Aprotinin, and 0.5mM PMSF). The lysate was incubated on ice with rocking for 15 min, filtered through miracloth and centrifuged at 4,000 x g for 5 min at 4˚C. After removing the cytoplasmic supernatant, the pellet containing intact nuclei was washed three times with cold A1 buffer. The pellet was re-suspended in 400 μl cold A1 buffer and protein concentration was determined by Protein Assay Dye Reagent Concentrate (Bio-Rad, Cat#500–0006) to ensure similar protein concentration of Day10 and 30 nuclei was used. The nuclear suspension was divided into four tubes and subjected to 5 min incubation at the following temperature: 25˚C (no heat-shock), 37˚C, 42˚C, and 50˚C. After additional 5 min incubation at 25˚C, the tubes were subjected to two cycles of freeze (liquid nitrogen)/thaw (25˚C) treatment. Half of the lysate from each tube was kept as "Total" fraction whereas the other half was centrifuged at 15,000 x g for 10 min at 4˚C. The supernatant collected after spin was labelled as "Extract" fraction. 6 x Laemmli buffer was added to "Total" and "Extract" fractions. The "Pellet" from the spin was resuspended in 1 x Laemmli buffer

such that the final volume of all three fractions is equal. Samples were boiled for 10 min and subjected to western analysis. The experiment was repeated 7 times and the relative level of Extract/Total lysate was calculated.

## Results and discussion

We hypothesized that epigenetic factors involved in mediating the onset of aging may undergo significant changes in their protein level during early adulthood. To test this possibility, *Drosophila* head and body lysates isolated at different adult ages were probed with antibodies against nuclear lamin-B, lamin-C, and several other nuclear proteins that are involved in genome organization. These include chromatin packaging histone H3 protein as well as nuclear architectural proteins like CP190 and CTCF. Male $w^{1118}$ flies have a median lifespan of 49 days (Fig 1A). Compared to Day 10 flies, the level of nuclear lamin-B protein shows modest yet statistically significant reduction in the heads as early as day 20 ($p < 0.0001$, Fig 1A; S1A Fig, S1 and S2 Tables in S1 File) and in the bodies at day 30 ($p < 0.01$, S1B and S1C Fig in S1 File). In contrast, the protein level of nuclear lamin-C (Fig 1B), histone H3, architectural protein CP190 and CTCF in the head tissues varied greatly among different biological replicates with no consistent pattern throughout aging (S2A Fig and S2 Table in S1 File). The reduction of lamin-B protein across different batches of 20 days old flies led us to further investigate if this pattern coincides with any measurable physiological decline in the animals. Negative geotaxis, an innate response where flies climb up the cylinder wall after being tapped to its bottom, senesces in flies [13]. Rapid iterative negative geotaxis assay (RING) showed that Day 20 and 30 males climbed about 2-fold slower than Day 10 males (Fig 1C and S3 Table in S1 File). Aside from reduced locomotor activity, Day 20 male flies also displayed lower resistance to both dry and wet starvation, as well as hydrogen peroxide and paraquat-induced oxidative stress than Day 10 flies (Fig 1D and S4 Table in S1 File). Therefore, reduced lamin-B protein level in Day 20 male fly heads correlates with their decline in locomotor activity and resistance to different environmental stresses. Same analyses were next performed using female flies (S2B–S2D Fig in S1 File). Although the level of lamin-B protein, locomotor activity and stress responses exhibit similar decline in early adulthood, Day 10 female flies succumbed to wet starvation more than Day 20 flies (S2D Fig in S1 File). Therefore, we focused our subsequent analysis only in male flies.

Mutations in *lamin-B* gene have been demonstrated to attenuate fly locomotor activity and shorten animal lifespan [14–16]. We reasoned if age-dependent reduction in lamin-B protein underlies the decline in animal physiology (Fig 1C and 1D), ectopic expression of *lamin-B* might rescue the decline in their locomotor activity and stress responses. To test this possibility, we generated transgenic male flies that expressed *UAS-Lamin-B-GFP* under the control of different *Gal4* promoters (S3A Fig in S1 File). Similar to previous report [15], ubiquitous expression of *lamin-B* using strong *Act5C* promoter led to pupal lethality with no eclosed adult fly (S3A Fig in S1 File). To circumvent this problem, *Hsp70-Gal4* promoter, which drives weak transgenic Gal4 expression in the absence of heat-shock treatment, was used to induce ubiquitous *lamin-B* expression (Fig 2A and 2B). Importantly, there is no leaky transgene expression from *UAS-Lamin-B-GFP/+ flies* (Fig 2B) and GFP-tag lamin-B (migrates at 100 kD) exhibits similar nuclear localization pattern as endogenous lamin-B protein (Fig 2C). Compared to control siblings, *lamin-B* overexpressing flies have improved locomotor activity of about ~1.3-fold (Fig 2D; S3B Fig and S3 Table in S1 File). Unexpectedly, transgenic flies with ubiquitous expression of *lamin-B* succumbed to starvation regiment as well as hydrogen peroxide and paraquat-induced oxidative stress as compared to their control siblings (Fig 2E; S3C Fig and S4 Table in S1 File). These results suggest that tissue-specific expression of *lamin-B* might

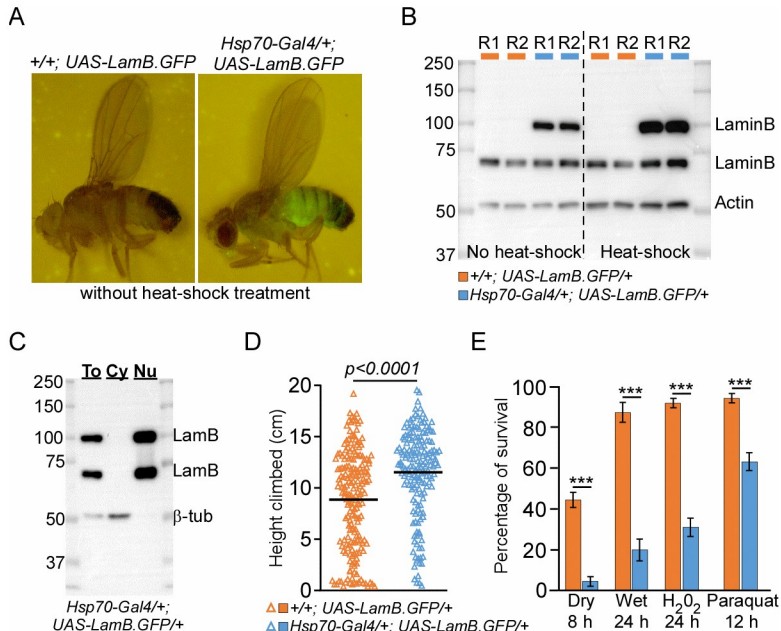

**Fig 2. *Hsp-70* induced expression of *lamin-B* improves locomotor activity but attenuates stress resistance.** (A) Images of *Hsp70-Gal4/+; UAS-LamB.GFP/+* and *+/+; UAS-LamB.GFP/+* flies in the absence of heat-shock treatment. (B) Immunoblot of untreated *Hsp70-Gal4/+; UAS-LamB.GFP/+* flies or flies harvested 24 hr after 90 min of 37°C heat-shock treatment. Five flies were loaded per lane. "R" stands for biological replicate. Endogenous lamin-B migrates around 70 kD whereas GFP-tag lamin-B runs at 100 kD. (C) Immuoblot of total lysate (To), cytoplasmic (Cy) and nuclear fraction (Nu) of male *Hsp70-Gal4/+; UAS-LamB.GFP/+* flies. (D) Locomotor activity and (E) stress tests of Day 30 non-heat-shocked *Hsp70-Gal4/+; UAS-LamB.GFP/+* and *+/+; UAS-LamB.GFP/+* male flies. Data presented as mean ± SEM. * $p < 0.002$, ** $p = 0.0003$, *** $p < 0.0001$.

be required to preserve fly locomotor activity without negatively impacting animal's physiology and lifespan.

Fly locomotor activity is controlled by diverse tissues such as the flight muscles [15] and dopaminergic neurons (DNs) within the protocerebral anterior medial (PAM) region of the brain [17]. Given that pan-neuronal knockdown of *lamin-B* with *elav4-Gal4* driver impairs fly motor function [16], we asked if there is any correlation between age-dependent decline in fly climbing ability and the lamin-B level in the DNs. To this end, we probed for endogenous lamin-B protein in Day 10 and 30 male *w^1118* fly brains, using tyrosine hydroxylase (TH) antibody to label DNs in the PAM region of the brain (Fig 3). Although the expression pattern of lamin-B in the DNs between the two aged groups was not statistically significant (Fig 3 and S5 Table in S1 File), the lower trend observed in Day 30 brains prompted us to examine the effect of *lamin-B* overexpression in PAM DNs. *R58E02-Gal4* promoter was chosen as it was known to label a large percentage of PAM DNs (Fig 4A) [17]. *TH-Gal4* and *DJ667-Gal4* [18] lines, which induced *lamin-B.GFP* expression in DNs outside of the PAM cluster (S4A Fig in S1 File) and muscles (S4B Fig in S1 File) respectively, were also used to test the functions of *lamin-B* in other cell-types. Flies with ectopic *lamin-B* expression in PAM DNs climbed about 1.3-times faster than their control siblings (Fig 4B; S4C Fig and S3 Table in S1 File). By contrast, ectopic *lamin-B* expression driven by *TH- and DJ667-Gal4* promoters caused significant reduction in their locomotor activity (S4D, S4E Fig and S3 Table in S1 File). Unlike earlier result with *Hsp70-Gal4* driver (Fig 2D), ectopic expression of *lamin-B* in PAM DNs by *R58E02-Gal4* promoter has modest or no impact on their responses to starvation and hydrogen peroxide-induced oxidative stress (Fig 4C).

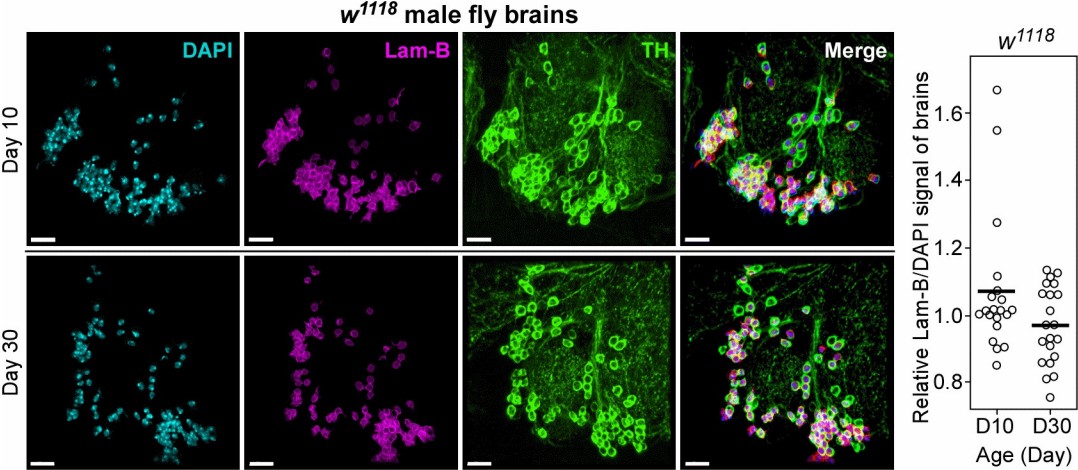

**Fig 3. Lower level of endogenous lamin-B in PAM DNs in Day 30 *w¹¹¹⁸* male fly brains.** Representative images and quantification of DNs within PAM regions of Day 10 and 30 brains stained with tyrosine hydroxylase (TH), nuclear lamin-B and DAPI. The scale is 10 μm, each circle represents a brain, and the black bar denotes the mean.

Overexpression of *lamin-B* in different tissues has been shown to shorten *Drosophila* lifespan [19]. Consistent with these findings, our results showed that flies expressing *lamin-B* under the control of *Hsp70-*, *TH-* and *DJ667-* promoters have reduced lifespan (S4F–S4H Fig and S6 Table in S1 File). However, we cannot rule out the possible toxicity induced by Gal4 protein as *Hsp70-Gal4/+* line has shorter lifespan as compared to +/+ and *UAS-LamB.GFP/+* genotypes (S4F Fig in S1 File). Interestingly, ectopic expression of *lamin-B* within the PAM DNs by *R58E02-Gal4* driver did not reduce animal lifespan (Fig 4D and S6 Table in S1 File). This supports the notion that tissue-specific manipulation of lamin-B level can promote animal fitness such as locomotor activity without impairing animal physiology or lifespan.

To better understand the mechanism underlying the gradual decline of lamin-B protein level during aging, we first examined its mRNA expression. The relatively stable mRNA level observed in aging male fly heads (Fig 5A) suggests that lamin-B protein might be regulated either post-transcriptionally or via post-translational modifications (PTMs). When lamin-B was immunoprecipitated (IP) from Day 10 and 30 flies, we observed a significant increase in the level of lamin-B migrated at around 100 kD in denaturing SDS gel (Fig 5B, red arrowhead). This suggests that subpopulation of lamin-B proteins might undergo specific PTMs during aging. In addition to PTMs, it is plausible that lamin-B may undergo other age-dependent biochemical changes. To this end, we examined the stability of lamin-B protein from Day 10 and 30 *w¹¹¹⁸* flies by cellular thermal shift assay (CETSA) where nuclei/cellular lysates heated at various temperatures were centrifuged into different fractions [12]. In this assay, proteins denatured by heating will precipitate whereas proteins that are chemically modified or protected by certain drugs/protein partners may remain soluble. Consistent with its role as structural scaffold protein at the nuclear matrix [11], majority of nuclear lamin-B protein from both age groups was found in the insoluble pellet (Fig 5C and S7 Table in S1 File). Surprisingly, 42˚C heat-shock treatment led to about 2.6-fold increase in the level of soluble lamin-B protein from Day 30 flies (Fig 5C and 5D). On the other hand, the level of soluble DNA-bound CP190 architectural protein from Day 30 flies was about 1.8-fold lower compared to Day 10 flies at different heat-shock temperatures (Fig 5C and 5D). These results indicate that both lamin-B and CP190 undergo significant changes to their thermal stability during aging in the flies.

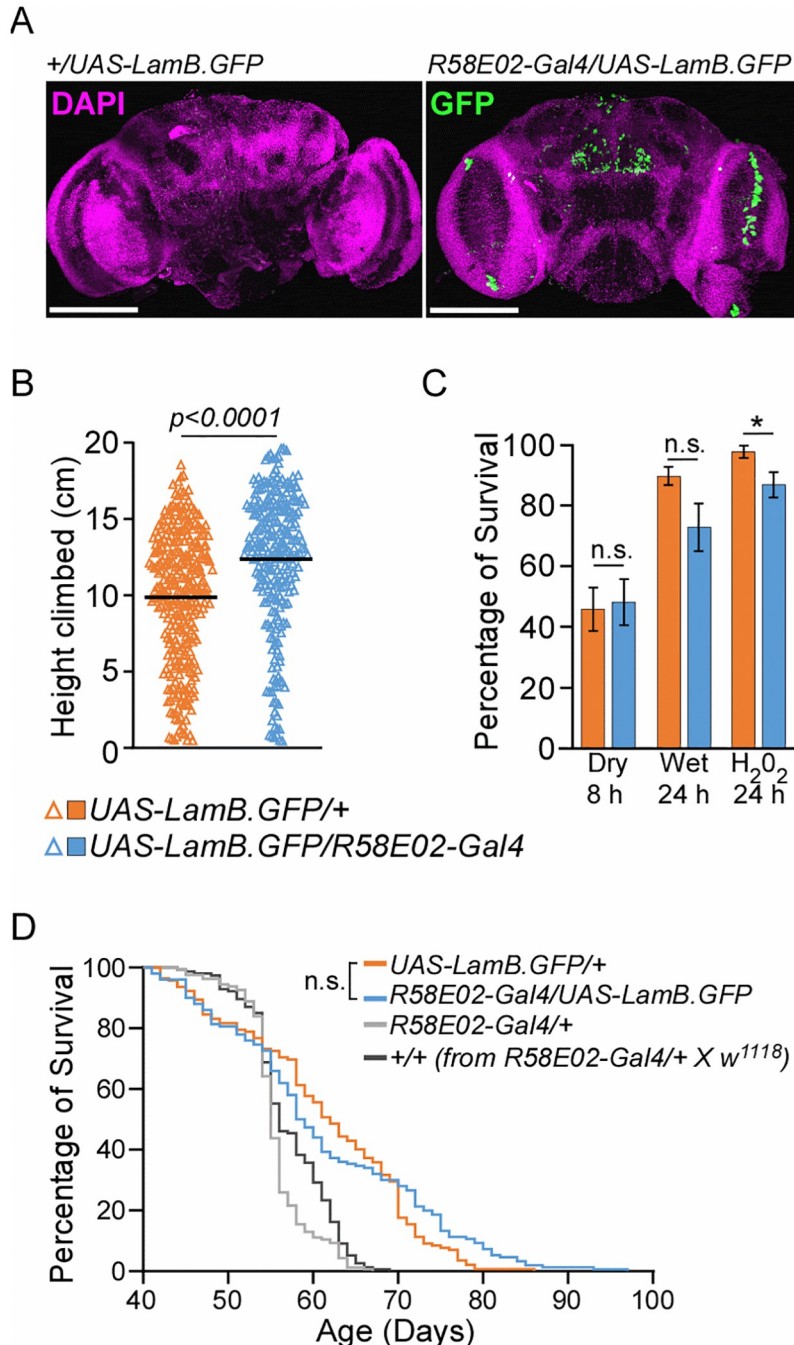

**Fig 4. Ectopic expression of lamin-B in PAM DNs improves locomotor activity without impairing stress resistance and lifespan.** (A) Expression pattern of GFP-tagged *Lamin-B* in the brains of male *R58E02-Gal4/ UAS-LamB.GFP* fly and its control sibling. Scale bars represent 150 μm. (B) Locomotor activity and (C) stress tests of Day 30 male *R58E02-Gal4/UAS-LamB.GFP* flies and their control siblings. Data presented as mean ± SEM. * *p* < 0.02. n.s.: non-significant. (D) Lifespan of male *R58E02-Gal4/UAS-LamB.GFP* flies and their different control siblings. Log-rank test.

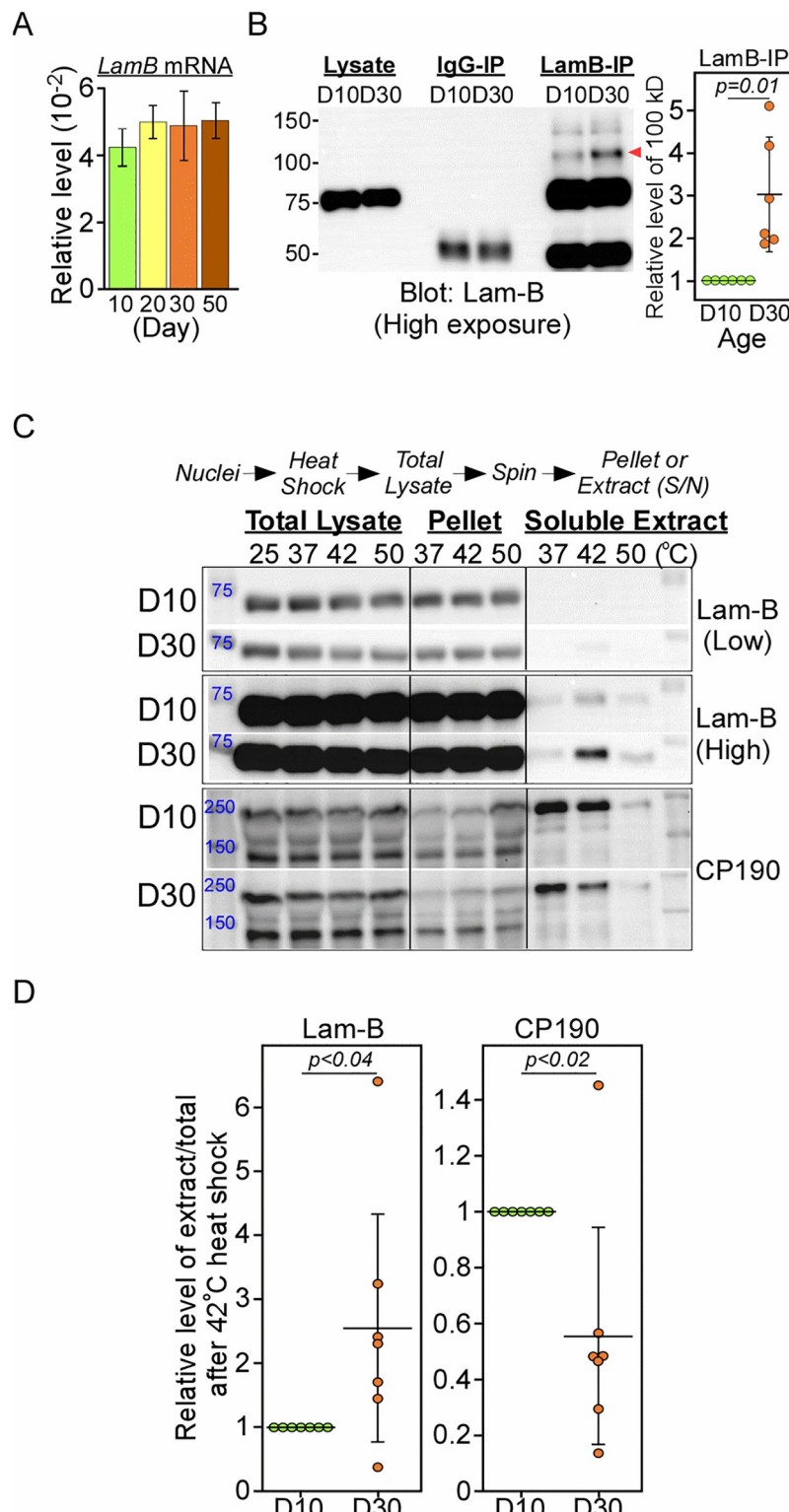

**Fig 5. Age-dependent changes in the thermal stability of nuclear lamin-B and CP190.** (A) No significant change in the level of lamin-B mRNA in $w^{1118}$ male fly heads at different ages (Day10-50). Data are presented as mean ± SEM; n = 3. (B) Western blot (Left) and quantification (Right) of nuclear lamin-B immunoprecipitation (LamB-IP) from Day10 and 30 male $w^{1118}$ flies. Data presented as mean ± SD, n = 6, paired t-test. (C) Western blot of different fractions of heat-shock nuclei from Day10 and 30 male flies. (D) Quantification of soluble nuclear lamin-B and CP190 protein after 42˚C heat-shock with data presented as mean ± SD, n = 7, paired t-test.

It is not fully clear what specific tissue or epigenetic signature may initiate the onset of physiological decline in animals like *Drosophila*. Here, we found that reduced level of lamin-B protein in the heads (Fig 1A) and DNs (Fig 3) correlates with the decline in the locomotor activity and stress responses of young male flies in their early adulthood. Ectopic expression of nuclear *lamin-B* protein in DNs within the PAM region of the brains improves male fly locomotor activity without compromising their stress resistance and lifespan. Taken together, these results suggest that reduced level of lamin-B protein within PAM DNs may initiate the onset of locomotor decline during aging. The level of lamin-B protein was highly variable across different biological replicates, mirroring the wide-range of locomotor activity and lifespan of individual fly observed in the otherwise genetically identical inbreed $w^{1118}$ population. Interestingly, recent reports showed that age-dependent downregulation of lamin B1 in adult neural stem/progenitor cells underlies age-related decline in mice hippocampal neurogenesis [20]. This data suggests that lamin-B1 may play a conserved role in maintaining neurological functions during aging.

The stable expression of *lamin-B* mRNA throughout fly lifespan suggests that its protein is likely to be regulated post-transcriptionally (Fig 5A). Consistent with this notion, IP experiments led to differential enrichment of 100 kD lamin-B protein in Day 30 male flies. One possible interpretation is that a small fraction of lamin-B (~5% of total ± 1 SEM, n = 6) undergoes specific PTMs and constant turnover, leading to its lower level in Day 30 male flies. Indeed, stability of lamin proteins has been demonstrated to be regulated by acetylation [21], methylation [22] and phosphorylation [23]. CETSA also revealed significant alterations in the thermal stability of nuclear lamin-B and architectural protein CP190 in Day 30 male flies. Lamin-B proteins form structural scaffolds that are highly insoluble. Surprisingly, heat-shock treatment at 42˚C led to increased solubility of lamin-B only in Day 30 male flies (Fig 5C). Although the mechanisms underlying such change only after 42˚C treatment remain unknown, this result highlighted the intrinsic difference of lamin-B and its potential interacting partners between Day 10 and Day 30 male flies. CP190 protein, which may also regulate fly locomotor activity [24], appears to have reduced solubility in Day 30 male flies. Given that lamin-B and CP190 regulate the formation of LADs at the nuclear periphery and topological associating domains of the chromosomes [5, 11], changes in their solubility are likely to impact genome reorganization and gene expression in Day 30 male flies. In addition, pathogenic tau has also been shown to alter nuclear lamina morphology in Alzheimer's disease [14]. The increased nuclear lamin-B solubility is likely to predispose nuclear lamina to tau-mediated changes in older animals. Future studies to uncover specific PTMs and binding partners of nuclear lamin-B will shed lights into the regulatory mechanisms that initiate the onset of aging.

## Supporting information

**S1 File.**
(PDF)

## Acknowledgments

The authors are appreciative of the assistance of intern Nancy Ly in the initial phase of the project.

## Author Contributions

**Conceptualization:** Chin-Tong Ong.

**Formal analysis:** Wei-Qi Lin, Tong-Wey Koh, Chin-Tong Ong.

**Funding acquisition:** Chin-Tong Ong.

**Investigation:** Wei-Qi Lin, Zhen-Kai Ngian, Chin-Tong Ong.

**Methodology:** Chin-Tong Ong.

**Project administration:** Wei-Qi Lin, Chin-Tong Ong.

**Supervision:** Chin-Tong Ong.

**Validation:** Wei-Qi Lin, Chin-Tong Ong.

**Writing – original draft:** Wei-Qi Lin, Chin-Tong Ong.

**Writing – review & editing:** Tong-Wey Koh, Chin-Tong Ong.

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
