## [Decision Letter · Decision Letter 0]

1 Dec 2021

PONE-D-21-29976Altered stability of nuclear lamin-B marks the onset of aging in DrosophilaPLOS ONE

Dear Dr. Ong,

Thank you for submitting your manuscript to PLOS ONE. After careful consideration, we feel that it has merit but does not fully meet PLOS ONE’s publication criteria as it currently stands. Therefore, we invite you to submit a revised version of the manuscript that addresses the points raised during the review process.

Although both reviewers  expressed favorable opinions about the manuscript, questions regarding experimental details were raised as outlined below.It is important that these points are thoroughly addressed, especially as it regards the male only versus sex independent conclusions of this study. This likely requires new experiments as outlined below. In addition, please incorporate the requested clarifications and corrections as detailed in the revised manuscript.

We look forward to receiving your revised manuscript.

Kind regards,

Efthimios M. C. Skoulakis, PhD

Academic Editor

PLOS ONE

Journal Requirements:

“The work is supported by Temasek Life Sciences Laboratory core funding 3160.”

“This work was supported by the core funding from Temasek Life Sciences Laboratory, Singapore.”

“The work is supported by Temasek Life Sciences Laboratory core funding 3160.”

Reviewers' comments:

Reviewer's Responses to Questions

**Comments to the Author**

1. Is the manuscript technically sound, and do the data support the conclusions?

Reviewer #1: Yes

Reviewer #2: Partly

2. Has the statistical analysis been performed appropriately and rigorously? 

Reviewer #1: Yes

Reviewer #2: Yes

3. Have the authors made all data underlying the findings in their manuscript fully available?

Reviewer #1: Yes

Reviewer #2: Yes

4. Is the manuscript presented in an intelligible fashion and written in standard English?

Reviewer #1: Yes

Reviewer #2: Yes

5. Review Comments to the Author

Reviewer #1: This a very well written and carefully conducted study showing a possibly important role of lamin-B in the onset of aging in Drosophila. The data have been collected and presented in a very solid and logical manner. The figures are nicely designed and informative. The text is balanced and the conclusions are correct without overinterpretation.

Minor comments:

- FigS2A: no clear motivation is provided about why specifically the proteins CP190, CTCF and Histone 3 were chosen.

- p12, line 12 (results section): pupae lethality > pupal lethality

- p14, line 26 (discussion section): mechanisms > the mechanisms; remains > remain

Reviewer #2: The authors demonstrated that the amount of Lamin-B protein content decreases upon aging in Drosophila adult males., contrarily to other proteins involved in nuclear organization of chromatin like CP190. This suggested a specific role of Lamin-B in the decline of neuronal and physiological functions during aging. The authors tested whether increasing the amount of Lamin-B could improve the functional decline, as well as modify the lifespan of adults. Using ubiquitous expression with the leakage expression level of a hsp70-Gal4 line, they authors show a rescue of the flies performance in the climbing assay, but no rescue of strong stresses like H2O2, paraquat…(there even is a decrease in survival)

Building on these data, they further show that increasing the level of expression of Lamin-B in specific dopaminergic neurons is responsible for the climbing assay rescue, and this does affect neither stress responses, nor lifespan.

To be able to further build on these results, I have several points I would like to be addressed:

- The study seems to have used male flies only. If this is the case, this should appear in the title (in Drosophila males), or I recommend 1° to precise, for each experiment, whether male flies or a mix of male and female flies was used, 2° to verify the obtained results, at least the main ones, with female flies. I can understand that using female bodies can lead to increased variability due to the egg content. However, this is not the case when working with female heads.

- In Fig. 2B, there is no decrease of Lam-B content at day 20, but there seems to be a decrease in the content of Lam-B GFP. Because the authors discuss about differences in solubility of the Lamin-B (and no change in the amount of expressed protein) during aging, it would be very interesting to see whether this is also the case for transgenic Lam-B GFP, which can easily be distinguished from Lam-B in western blots.

- Also, the authors do not present the amount of Lam-B GFP in UAS-lam-B GFP/+ flies. We know that UAS promoter can also be leaky depending on the genomic insertion site, and it is possible that there is already expression of Lam-B GFP in this genotype, in particular considering the rescue activity presented in Fig. S2 and Fig. S3. Western blots should be shown about this (The GFP fluorescence presented in Fig. 2A is not enough, because it is not possible to detect low levels of fluorescence).

- Also, to confirm 1) that Lam-B-GFP behaves like endogenous Lam-B (same localization into protein complexes, same behavior during aging), the authors could take advantage of the UAS-Lam-B-GFP line (w/o any hsp70 or other Gal4 line) to test for solubility of this protein at different temperatures like in Fig 5C. 2) This would also strengthen their data about changes in solubility (or PTM / or interactome) of Lam-B during aging.

- What are the data for +/+ flies or hsp70-Gal4/+ flies in the stress experiments in the conditions tested in Fig. 2 (8h, 12h and 24h ?).

- In all lifespan data presented, the longest lifespan is for UAS-LamB-GFP, suggesting again some leakage of expression from this stock (see point 3). Testing this hypothesis would allow a better interpretation of the author’s results.

6. PLOS authors have the option to publish the peer review history of their article (what does this mean?). If published, this will include your full peer review and any attached files.

Reviewer #1: No

Reviewer #2: No

---

## [Author Response · Author response to Decision Letter 0]

10 Feb 2022

1. The original uncropped and unadjusted immunoblot images were included in Section (8) of Supporting Information.

2. Funding Statement: The work is supported by Temasek Life Sciences Laboratory core funding 3160. The funders had no role in study design, data collection and analysis, decision to publish, or preparation of the manuscript.

3. The phrase “data not shown” was previously used to describe pupa lethality. In this revision, the numbers of eclosed adult flies were presented in Fig.S3A, page 6 of Supporting information. There was 0 Actin-Gal4/UAS-LamB-GFP fly as compared to 98 males and 78 females UAS-LamB-GFP/Tm6B flies.

4. We have also performed all the experiments suggested by the reviewers. See details in response to reviewers.

---

## [Editor Report · Decision Letter 1]

28 Feb 2022

Altered stability of nuclear lamin-B marks the onset of aging in male Drosophila

PONE-D-21-29976R1

Dear Dr. Ong,

We’re pleased to inform you that your manuscript has been judged scientifically suitable for publication and will be formally accepted for publication once it meets all outstanding technical requirements.

Kind regards,

Efthimios M. C. Skoulakis, PhD

Academic Editor

PLOS ONE
---

## [Editor Report · Acceptance letter]

17 Mar 2022

PONE-D-21-29976R1 

Altered stability of nuclear lamin-B marks the onset of aging in male *Drosophila*

Dear Dr. Ong:

I'm pleased to inform you that your manuscript has been deemed suitable for publication in PLOS ONE. Congratulations! Your manuscript is now with our production department. 

Kind regards, 

on behalf of

Dr. Efthimios M. C. Skoulakis 

Academic Editor

PLOS ONE